# From Steps to Context: Optimizing Digital Phenotyping for Physical Activity Monitoring in Older Adults by Integrating Wearable Data and Ecological Momentary Assessment

**DOI:** 10.3390/s25030858

**Published:** 2025-01-31

**Authors:** Kim Daniels, Kirsten Quadflieg, Jolien Robijns, Jochen De Vry, Hans Van Alphen, Robbe Van Beers, Britt Sourbron, Anaïs Vanbuel, Siebe Meekers, Marlies Mattheeussen, Annemie Spooren, Dominique Hansen, Bruno Bonnechère

**Affiliations:** 1Centre of Expertise in Care Innovation, Department of PXL—Healthcare, PXL University of Applied Sciences and Arts, 3500 Hasselt, Belgium; kirsten.quadflieg@pxl.be (K.Q.); jolien.robijns@pxl.be (J.R.); hans@beweegmeer.be (H.V.A.); annemie.spooren@uhasselt.be (A.S.); bruno.bonnechere@uhasselt.be (B.B.); 2REVAL Rehabilitation Research Center, Faculty of Rehabilitation Sciences, Hasselt University, 3590 Diepenbeek, Belgium; robbe.vanbeers@student.uhasselt.be (R.V.B.); britt.sourbron@student.uhasselt.be (B.S.); anais.vanbuel@student.uhasselt.be (A.V.); siebe.meekers@student.uhasselt.be (S.M.); marlies.mattheeussen@student.uhasselt.be (M.M.); dominique.hansen@uhasselt.be (D.H.); 3PXL Research, Centre of Expertise in Smart-ICT, PXL University of Applied Sciences and Arts, 3500 Hasselt, Belgium; jochen.devry@pxl.be; 4BIOMED Biomedical Research Instititute, Faculty of Medicine and Life Sciences, Hasselt University, 3590 Diepenbeek, Belgium; 5Technology-Supported and Data-Driven Rehabilitation, Data Sciences Institute, Hasselt University, 3590 Diepenbeek, Belgium

**Keywords:** digital phenotyping, ecological momentary assessment, physical activity patterns, wearable, aging

## Abstract

Physical activity (PA) is essential for healthy aging, but its accurate assessment in older adults remains challenging due to the limitations and biases of traditional clinical assessment. Mobile technologies and wearable sensors offer a more ecological, less biased alternative for evaluating PA in this population. This study aimed to optimize digital phenotyping strategies for assessing PA patterns in older adults, by integrating ecological momentary assessment (EMA) and continuous wearable sensor data collection. Over two weeks, 108 community-dwelling older adults provided real-time EMA responses while their PA was continuously monitored using Garmin Vivo 5 sensors. The combined approach proved feasible, with 67.2% adherence to EMA prompts, consistent across time points (morning: 68.1%; evening: 65.4%). PA predominantly occurred at low (51.4%) and moderate (46.2%) intensities, with midday activity peaks. Motivation and self-efficacy were significantly associated with low-intensity PA (R = 0.20 and 0.14 respectively), particularly in the morning. However, discrepancies between objective step counts and self-reported PA measures, which showed no correlation (R = −0.026, *p* = 0.65), highlight the complementary value of subjective and objective data sources. These findings support integrating EMA, wearable sensors, and temporal frameworks to enhance PA assessment, offering precise insights for personalized, time-sensitive interventions to promote PA.

## 1. Introduction

Physical activity (PA) is widely recognized as a cornerstone of public health, playing a pivotal role in reducing the burden of non-communicable diseases and enhancing well-being across the lifespan [1,2]. However, the accurate and multidimensional assessment of PA, particularly among older adults, remains a pressing challenge that limits the effectiveness of intervention strategies and public health initiatives. Accurate measurement is essential for understanding its determinants, monitoring trends, and developing effective interventions [3,4]. Historically, PA research relied heavily on retrospective self-report questionnaires, which, despite offering large-scale population insights, were prone to recall bias, inaccuracies, and a limited capacity to capture nuanced, time-sensitive behaviors [5,6,7]. These methodological limitations restricted the field’s ability to explore the dynamic and contextual nature of PA and its relationship with behavioral and physiological determinants [8,9].

The introduction of objective tools such as pedometers, accelerometers, and wearable devices marked a significant turning point in PA research [10]. These tools offer reliable, precise, and reproducible data on the frequency, intensity, and duration of PA, facilitating greater standardization and reproducibility [11,12,13]. However, these tools often lack the ability to capture contextual and psychosocial factors, creating a gap between the collected data and actionable insights required for effective interventions [14,15,16]. To bridge this gap, digital phenotyping that leverages continuous, real-world data collection through personal devices such as smartphones and wearables has emerged as a transformative methodology [17,18,19,20,21,22]. By capturing multidimensional data, digital phenotyping enables deeper exploration of PA behaviors, physiological signals, and contextual influences, providing insights into temporal and situational patterns that traditional tools often miss [19].

The ability to integrate diverse metrics, such as heart rate, steps, and environmental conditions, has expanded the scope of PA research, enabling a more holistic understanding of its biopsychosocial determinants [23]. Despite these advancements, there remains limited understanding of how these data streams can be optimized and tailored to specific populations, such as older adults, who face unique challenges related to mobility, health conditions, and motivational barriers [24,25,26].

Moreover, despite these innovations, the field continues to face critical methodological gaps. Many studies still rely on cross-sectional designs emphasizing between-subject variability, offering valuable but incomplete insights [27]. For instance, such analyses reveal population-level associations such as the relationship between higher PA levels and better cardiovascular profiles, yet they fail to capture the intra-individual factors driving PA engagement or avoidance. This limitation is particularly pronounced when examining behaviors that fluctuate across time and contexts within the same individual [28]. Moreover, the ecological fallacy—where between-subject findings are wrongly generalized to within-subject dynamics—underscores the need for more sophisticated analytical approaches [4,29,30]. These discrepancies highlight the necessity of capturing within-subject variations to develop actionable insights [31,32]. To address these limitations, intensive longitudinal data collection through ecological momentary assessment (EMA) offers a promising approach [33,34]. EMA facilitates high-frequency, real-time data collection of behaviors, emotions, and contexts in naturalistic settings, capturing time-dependent variations that static methods cannot [35,36]. This methodology provides a granular view of PA behavior, enabling researchers to understand why individuals engage in or avoid PA at specific times or in particular circumstances. Such within-subject insights are invaluable for designing interventions that address individual-level barriers, such as adapting routines, providing motivational feedback, or modifying environmental factors. By focusing on dynamic behavioral patterns, EMA extends beyond static population averages to reveal the real-world drivers of PA [37,38,39,40].

However, questions remain regarding optimizing these methodologies for specific populations, particularly older adults. Age-related changes in behavior, physiology, and adoption of technology present unique challenges that require tailored solutions [41,42,43]. This study addresses this critical gap by examining how digital phenotyping methodologies, specifically the integration of EMA and wearable sensor data, can provide nuanced, within-subject insights into PA patterns among older adults. The overarching aim is to determine how these methodologies can balance comprehensive data collection with participant feasibility while generating actionable insights into PA behaviors. To this end, this study employed a novel combination of real-time EMA and wearable data collection, with the following aims:Assessing the interplay between PA patterns and psychosocial determinants across different times of the day and intensities;Evaluating the feasibility of these methodologies and participant adherence within a representative sample of older adults;Exploring optimal durations of data collection to ensure participant engagement while capturing meaningful behavioral trends.

Focusing on these objectives, this research aims to advance the methodological toolkit available for PA research, providing a foundation for personalized interventions and scalable public health solutions. Ultimately, the findings contribute to bridging the gap between technological advancements and practical applications in promoting PA among older adults, contributing to healthier aging and improved quality of life.

## 2. Materials and Methods

### 2.1. Study Setting and Design

This study employed a two-week prospective observational design that integrated both supervised and unsupervised methods to provide comprehensive, multidimensional insights into PA behaviors and their determinants, as illustrated in Figure 1. The supervised components included self-reported assessments, which collect demographic and contextual information, while unsupervised data collection leveraged EMA and continuous real-world monitoring through a wearable device. The two-week period was chosen based on existing literature on EMA [33,44,45] and PA monitoring [46,47,48,49] with wearables, which indicates that at least four to seven days are required to capture variability in daily behaviors, including both weekdays and weekends. To enhance data reliability and ensure all potential fluctuations due to contextual and environmental factors were accounted for, we extended the observation period to two weeks. This approach provided a more comprehensive view of participants’ PA patterns while maintaining feasibility and minimizing participant burden. The study was conducted in a naturalistic setting to ensure ecological validity, allowing participants to engage in their daily routines without interference.

This study was registered at Clinical Trials.gov (NCT06094374) on 17 October 2023 and approved by the Ethical Committee of Hasselt University (B1152023000011). The full study protocol detailing recruitment strategies, data collection procedures, and analytical methods has been presented separately [50]. Informed consent was obtained from all subjects before participation (Section A.1).

### 2.2. Participants

This study recruited older adults aged 65 and older, targeting community-dwelling individuals living independently at home or in serviced apartments. Participants were required to be competent to provide informed consent, able to actively participate in the study, and free from severe illnesses that could impair mobility, functional capacity, or cognitive ability to the extent that they could not comprehend or follow instructions. The inclusion and exclusion criteria (detailed in Section A.2) ensured that participants were native Dutch speakers without current neurological, cardiovascular, respiratory, severe metabolic, or cognitive disorders. Importantly, digital literacy was not a criterion for exclusion, allowing for a diverse range of participants with varying levels of digital competence. Recruitment occurred between February and September 2024 through a combination of online and offline strategies. These included social media outreach, newspaper advertisements, presentations at senior citizen organizations, and collaboration with local community services. Because of the lack of accessible prior studies that could provide foundational information, sample size calculation was impossible. Therefore, a convenient sample of at least 100 healthy older adults was selected for this trial [19]. We performed interim analyses at regular milestones, such as after recruiting 40, 80, 100 participants, and so on. During these interim analyses, if we had determined that the data had reached a point of saturation—indicating that additional participants were unlikely to produce significantly different findings—the sample size would have been finalized. A comprehensive CONSORT flowchart (Section B.2) outlines the recruitment process and participant flow.

### 2.3. Data Collection

#### 2.3.1. Baseline Assessment

Self-reported demographic and socio-economic data including age, sex, marital status, education level, living arrangements, and previous incidence of falls were collected [51,52]. PA was assessed with the International Physical Activity Questionnaire—short form (IPAQ) [53,54,55,56,57,58,59]. Digital readiness of the participants was measured using the Digital Health Readiness Questionnaire (DHRQ) [60]. These data were collected via the online survey platform Qualtrics [61].

#### 2.3.2. Physical Activity Monitoring

Throughout the two-week trial, participants’ daily activities were continuously monitored using the Garmin Vivosmart 5^®^ activity tracker (Garmin International, Olathe, KS, USA), which provided 24/7 data collection. This wearable device recorded a range of parameters, including stress levels, physical activity (PA), step count, calorie expenditure, heart rate, floors climbed, physical activity intensity, cardiometabolic metrics, body battery, and sleep patterns. Walking cadence served as a reliable indicator of PA intensity. Moderate intensity was characterized by activity exceeding 3 metabolic equivalents (METs), equating to a cadence of at least 100 steps per minute. Light intensity PA fell within the range of 1.6 to 2.9 METs, while movements below 20 steps per minute were classified as incidental and categorized as sedentary behavior. The findings were reported in terms of minutes per week, aligning with the World Health Organization’s recommendations on the minimum required levels of physical activity [62]. The Garmin wearable enabled non-intrusive, real-time tracking, offering a comprehensive view of participants’ routines and health-related behaviors over the trial period. The Garmin Vivosmart 5^®^ proved to be a valid measurement tool for older adults [63,64,65].

#### 2.3.3. EMA

Participants engaged via the SEMA3 smartphone application [66] (Melbourne eResearch Group, Melbourne, Australia) receiving four randomized prompts daily between 8:00 AM and 11:00 PM. Prompts were evenly distributed across four-time intervals: 8:00–11:00, 12:00–15:00, 15:00–18:00, and 18:00–23:00. Participants were instructed to respond immediately to each prompt (completion time: 2–3 min with an expiration time of 30 min) unless engaged in incompatible activities, such as driving. Non-responses triggered up to three reminders at 5-min intervals, after which access to the EMA questionnaire was suspended until the next scheduled prompt [66].

Participants were asked to assess five key domains: physical well-being, mental well-being, motivation, efficacy, and context, on a 7-point Likert scale. These assessments included self-rated health, physical complaints such as muscle stiffness, pain, dizziness, shortness of breath, and fatigue, as well as contextual factors and overall quality of life (QoL). To minimize response bias and enhance data reliability, the questionnaire items were presented in a randomized order [31].

#### 2.3.4. Follow-Up Duration Analysis

To determine the optimal duration for data collection, data from varying periods (e.g., three, seven, ten, and 14 days) were compared. This analysis aimed to identify the minimum duration necessary to capture meaningful behavioral trends without compromising participant engagement. Daily PA patterns were analyzed by intensity levels (low, moderate, vigorous) for each timeframe to assess whether shorter durations adequately reflected overall activity levels. Similarly, EMA adherence rates were evaluated across the different timeframes to understand the impact of follow-up duration on participant responsiveness and engagement. Finally, the correlation between subjective measures (e.g., motivation and self-efficacy from EMA) and objective PA metrics (e.g., step counts) was compared to determine whether meaningful relationships could be identified in shorter observation windows.

### 2.4. Data Processing

To analyze activity patterns with greater granularity, the Garmin data were divided into five distinct time periods throughout the day: night, morning, noon, afternoon, and evening, the latter four corresponding to the time period for EMA evaluation. This categorization enabled detailed temporal analysis of PA levels throughout the day, including integration with EMA data. PA was categorized into three intensity levels for these different periods: low, moderate, and vigorous. These classifications were based on the device’s built-in algorithms for activity recognition, with particular attention to walking activities. Data completeness was assessed by examining the number of recorded measurements per participant daily. Only days with complete 24-h recordings were included in the analysis.

Concerning the EMA, data processing involved several stages to ensure the dataset was prepared for accurate and meaningful statistical analysis. Our methodology, implemented in R, focused on transforming and categorizing the data, filtering out irrelevant information, and aggregating key metrics. The timestamp data were parsed to extract date and time information. Days were categorized as either weekday or weekend. The time of day was classified into four distinct periods according to when the data were collected. Aggregated scores were calculated for physical health (6 items), mental health (8 items), motivation (4 items), efficacy (2 items), and context (2 items) by summing individual item responses. To simplify the comparison between the categories, the scores were adjusted according to the number of questions asked in each dimension for each category. These composite scores facilitated a holistic analysis of each construct. Response adherence was calculated as the percentage of completed assessments relative to the total number of prompted assessments.

Finally, we merged the Garmin-derived step-count data with the EMA responses using participant ID, day number, and time category as matching variables to examine the relationship between PA patterns and EMA responses. The integrated dataset included step counts categorized by PA intensity (low, moderate, vigorous) and five EMA dimensions: physical health, mental health, motivation, self-efficacy, and context.

### 2.5. Statistical Analysis

Multiple analytical and visualization techniques were employed to examine the relationships between PA and EMA responses.

To visualize the period of most intensive activity within the day, we plotted the aggregated number of steps per intensity on a clock diagram. We also recorded the most active period of the day, defined as the 15 min with the highest number of steps, for the different intensities. We plotted this to assess the stability of this parameter over time.

For the EMA, temporal analysis of adherence patterns was conducted to examine potential trends over the study period and across different times of day. Linear mixed-effects models were employed to analyze the temporal patterns in the EMA responses, with participant and assessment type treated as random effects. The models accounted for both the nested structure of the data (multiple observations per participant) and the potential interaction between the time of day and study progression (days). Model selection was performed using likelihood ratio tests and AIC comparisons.

Correlation analyses were performed using Pearson’s method with false discovery rate (FDR) correction for multiple comparisons. The relationships between variables were visualized using correlation matrices and heatmaps, with data standardization applied to account for different measurement scales. The analysis was stratified by PA intensity level to examine whether the relationships between step counts and EMA measures varied across different activity intensities. Additionally, temporal patterns were investigated by analyzing these relationships across different times of day, controlling for potential time-of-day effects on both PA and psychological states.

For multivariate visualization, a hierarchical clustered heatmap was generated using the standardized scores for six variables (steps and five EMA dimensions), with data split by PA intensity level. This integrated approach ensured a thorough and insightful analysis of the combined Garmin and EMA data, facilitating a deeper understanding of the relationships between PA and mental health metrics.

Finally, to determine the optimal duration of the follow-up, we first analyzed the different outcomes (e.g., EMA, Garmin, correlation) during the full follow-up (14 days). Using a random sliding window, we segmented the 14-day dataset into overlapping subsets of varying lengths (3, 7, 10, 12, and 14 days). For each window length, subsets were generated by randomly selecting starting points within the follow-up period, ensuring diverse coverage of the data. The median values of the outcomes within these windows were then computed to provide a representative summary for each duration. This method reduced biases that might have arisen from analyzing only fixed or pre-determined intervals and allowed robust comparison across different follow-up durations. The Kruskal–Wallis test was subsequently applied to evaluate differences in the median values across these durations, offering a non-parametric assessment of the optimal follow-up period.

All statistical analyses were conducted using R packages, including lme4 for mixed-effects modeling and ggplot2 for visualization. Multiple comparison corrections were applied using the FDR method where appropriate.

## 3. Results

### 3.1. Baseline Assessment: Participants’ Demographic Characteristics

The study included 108 participants with a median age of 69, ranging from 65 to 87. Among the participants, 60 were female (55.6%), and the median BMI was 26.3 kg/m^2^, with values ranging from 19 to 42. Regarding education, nearly half of the participants had completed high school (48.2%, n = 52), with others reporting secondary education (25.9%, n = 28), university degrees (21.3%, n = 23), primary education (2.8%, n = 3), or PhDs (1.8%, n = 2).

PA levels assessed through the IPAQ-SF revealed a median total MET-min/week of 5154 (IQR = 7331), with 77 participants (71.3%) reporting high activity levels, 30 (28%) reporting moderate activity, and only 1 (0.7%) classified as having low activity. Digital literacy scores were relatively high, with a mean total score of 60 ± 8 out of 75. The complete characteristics of the participants are presented in Table 1.

### 3.2. Physical Activity

#### 3.2.1. Relationship Between Self-Reported Outcome Measures of PA and Objective Measures

As presented in Figure 2, our analysis revealed a total absence of correlation between the total number of steps per day and the self-reported MET values (R = −0.026, *p* = 0.65). The same conclusion was found when performing subgroup analysis at the different intensity levels: low (R = 0.07, *p* = 0.45, moderate (R = −0.10, *p* = 0.32), and vigorous (R = −0.09, *p* = 0.36).

#### 3.2.2. Temporal Patterns of Physical Activity: Analysis of Daily Patterns and Weekly Variations in Step Distribution and Intensity Level

We analyzed the total number of steps per day according to the three intensity levels. As shown in Figure 3, significant differences were observed in step distribution across intensity levels (*p* < 0.001, chi^2^ test). The majority of steps were performed at low (51.4%) and moderate (46.2%) intensity levels, with only 2.4% of steps classified as vigorous intensity. Notably, this distribution was not influenced by the day of the week (*p* = 0.89, chi^2^ test). As presented in Figure 3, the median number of steps was lower than the 10,000 steps recommended by the WHO. In order to analyze this in more detail, we computed the number of participants that reached this threshold every day. For the 1512 days analyzed, only 454 (30%) reached this threshold. Figure 4 shows the repartition of the patients according to the number of days they reached the threshold. Notably, none of the participants reached the recommended number of steps per day during the entire follow-up period, and 15 (14%) participants did not reach it on any given day.

When examining the overall number of steps per day, we found a statistically significant difference (*p* < 0.001) between weekdays and weekends, as illustrated in Figure 5. However, when steps were compared individually across the seven days of the week, no statistically significant difference was identified (*p* = 0.47, Kruskal–Wallis test). Similarly, no significant differences were found in the total number of steps by intensity level for low (*p* = 0.35), moderate (*p* = 0.64), or vigorous intensity (*p* = 0.74).

To obtain more insight into the temporal distribution of PA, we plotted the aggregated number of steps, at the different intensity levels, on a watch (Figure 6). We observed that most of the activity occurred in the morning at low and moderate intensity, as already noted. The vigorous activity, while being significantly limited, mostly occurred exclusively in the morning. The individual results are plotted in Figure 7, highlighting the huge variability not only in terms of PA levels (i.e., total number of steps) but also in terms of patterns and time preferences.

However, despite this high between-subject variability, PA patterns demonstrated consistent temporal rhythms over the course of the 14-day period, as illustrated in Figure 8, which represents the most active hour of the day for the different intensities through the course of the follow-up. The most active hour of the day, regardless of the intensity, was between 11:00 and 12:00. No difference was found between the intensities nor the effect of the day (β = −0.09 (SE = 0.01), *p* = 0.65).

### 3.3. EMA Results

#### 3.3.1. Adherence

The overall adherence rate was 67.2% (SD = 8.4) across the entire follow-up period (Figure 9). No statistically significant differences were found between time periods (*p* = 0.82), with adherence rates of 67.4% (7.3) in the morning, 68.2% (10.0) at noon, 68.0% (7.8) in the afternoon, and 65.4% (8.8) in the evening.

#### 3.3.2. Correlations Between Dimensions of EMA

Efficacy and context exhibited the strongest positive correlation (r = 0.62, *p* < 0.001), indicating a close relationship between self-efficacy and situational or environmental factors. Context also demonstrated moderate positive associations with efficacy (r = 0.51, *p* < 0.001) and motivation (r = 0.38, *p* < 0.001), underscoring the role of environmental influences on self-efficacy and drive. Mental health was moderately correlated with physical health (r = 0.42, *p* < 0.001), reflecting the interconnectedness between mental and physical states.

#### 3.3.3. Temporal and Intraday Variability in EMA and PA

A linear mixed-effects model was employed to analyze the interaction between time of day and day on EMA responses. This model accounted for the nested random effects of repeated EMA measurements on participants, using maximum likelihood estimation. The analysis revealed significant main effects for time of day (β = −2.54, SE = 0.27, t = −9.52) and day (β = −0.44, SE = 0.09, t = −4.84), as well as a significant interaction between time of day and day (β = 0.16, SE = 0.03, t = 4.96). The random effects structure indicated substantial variability between participants (SD = 44.14) and across EMA sessions (SD = 14.04).

Response scores across the five EMA categories demonstrated distinct patterns throughout the day, highlighting variations in participant engagement and response consistency across morning, noon, afternoon, and evening (Figure 10A). Efficacy consistently showed the highest response scores, remaining stable during the earlier parts of the day but exhibiting a slight decline in the evening. Context followed a similar trend, maintaining high scores with a modest decrease as the day progressed. The categories of motivation and mental health presented moderate response scores that declined slightly from morning to evening. Physical health displayed the lowest and most stable daily scores, with minimal variability. The general downward trend across most categories, particularly efficacy and motivation, suggests a potential diurnal effect, with greater engagement and accuracy earlier in the day. Figure 10B depicts the evolution of PA throughout the day, categorized into low, moderate, and vigorous intensities. Low-intensity activity was the most prevalent across all time periods, showing relatively steady levels from morning to evening, though with considerable variability as indicated by the wide error bars. Moderate-intensity activity peaked around noon before declining through the afternoon and evening. In contrast, vigorous activity remained consistently low throughout the day, slightly increasing at noon but involving fewer steps than low and moderate activities. Regardless of intensity, PA tended to be highest at noon and declined sharply by evening, suggesting reduced movement as the day progressed.

### 3.4. Integrated Model

To better understand the relationship between EMA variables—context, efficacy, motivation, mental health, physical health—and the PA monitored by the wearable, we performed correlation analysis. First, we evaluated correlations for the overall number of steps per day, regardless of the time of the day. Statistically significant positive correlations were found between PA and motivation (R = 0.16, *p* = 0.001), context (R = 0.11, *p* = 0.007), efficacy (R = 0.10, *p* = 0.006), and mental health (R = 0.09, *p* = 0.031) but not for physical health (R = −0.05, *p* = 0.31).

Since important variations in both PA and EMA were observed throughout the day (Figure 5 and Figure 10), we further analyzed these relationships according to the time of the day (Figure 11). For the mornings, we still found a positive and statistically significant correlation between mental health and PA (R = 0.26, *p* = 0.008); at noon, there was a positive and statistically significant correlation between motivation and PA (R = 0.22, *p* = 0.046). No statistically significant correlations were found between PA and EMA in the afternoon or evening.

Interestingly, when comparing these correlations according to the intensity levels, we observed statistically significant correlations only at low levels of PA for motivation (R = 0.20, *p* = 0.003), efficacy (R = 0.14, *p* = 0.013), and context (R = 0.13, *p* = 0.019). Step counts remained totally uncorrelated with the psychological variables across all time periods, indicating a minimal direct relationship between PA levels and subjective psychological constructs.

The heatmap visualized in Figure 12 reveals relationships between the EMA categories and PA across the three activity levels: low, moderate, and vigorous. The clustering of participants and variables highlights patterns within and across activity levels. In the low-activity group, variability is observed, with mental health and motivation showing fluctuating patterns, while physical health and context remain more stable. More structured clustering emerges in the moderate activity group, particularly around efficacy and context, indicating stronger links between environmental factors and physical states. In the vigorous activity group, clustering becomes highly localized, with a relationship between high physical health and motivation.

We performed a hierarchical cluster analysis of PA and EMA for the different times of day (Figure 13). Motivation and self-efficacy exhibited strong positive associations, largely irrespective of the time of day. Context appeared to be negatively associated with these factors. PA showed a moderate positive association with mental state, particularly in the afternoon cluster.

Both heatmaps show consistent clustering of self-efficacy and motivation, suggesting a strong positive correlation. Context appears to have been inversely related to these factors, particularly in the moderate and low-intensity PA clusters. Lastly, PA showed some positive association with mental state, especially in the afternoon and vigorous activity clusters.

### 3.5. Optimal Following Time

Since one of our research questions was to determine the optimal follow-up duration, we compared our results both for PA and EMA for different windows of time, namely across 14 days (complete duration), one random day, three consecutive random days, seven consecutive random days, ten consecutive random days, and twelve consecutive random days.

The complete results are presented in Table 2. PA levels remained consistent across all durations and intensities, with no significant differences observed for the different intensity levels. Similarly, no statistically significant difference was found between EMA values for the different time windows.

### 3.6. Interplay Between PA Monitoring and EMA

Finally, Table 3 summarizes the findings addressing the study’s research goals, highlighting the relationship between PA patterns, psychosocial determinants, and adherence to the methodologies employed. It provides detailed insights into temporal activity patterns, adherence rates, and optimal durations for data collection, offering a comprehensive understanding of participant engagement and the feasibility of the study design. Each research goal is addressed based on the detailed analysis and statistical outcomes presented in the results section.

## 4. Discussion

This study employed EMA and wearable devices to conduct a comprehensive, objective analysis of PA behaviors and psychological states among older adults.

The results demonstrated no correlation between self-reported METs, as measured by IPAQ-SF, and objective step counts recorded by wearable devices. This lack of correlation was particularly evident at higher PA intensities, aligning with previous research highlighting the limitations and inaccuracies of self-reported measures in capturing physical activity levels [67,68]. Comparison between steps per day according to the wearable and self-reported METs from the IPAQ was included to explore potential discrepancies between objective and self-reported activity measures, emphasizing the inherent challenges in accurately capturing intensity and volume through self-reports. Although there was no direct conversion between steps and self-reported METs, their inclusion enabled examination of different dimensions of PA behavior, as supported by previous studies [67,69]. The absence of correlation between these measures can be attributed to several factors. First, recall bias and social desirability are known to contribute to the underreporting of vigorous activities and the overestimation of lighter-intensity activities [70,71,72,73]. Moreover, participants may overestimate their activity levels in self-reports due to a desire to present themselves in a socially favorable light, inflating MET values [74,75,76]. Similarly, older adults may face challenges estimating activity intensity and duration due to cognitive biases, further compounding these discrepancies [77]. Secondly, self-reported MET values also differ in measurement scope compared with wearable devices. While wearable devices capture granular, continuous data primarily focused on ambulatory activities, MET values also account for a broader range of activity types such as cycling or swimming, which are not directly recorded by step counters [46,78,79,80]. This disparity can contribute to variability in correlation. Third, wearables often use algorithms to classify activity intensity, which may not align with participants’ perceptions reported in self-assessments [81]. Lastly, variability in device sensitivity and activity types adds complexity. For example, wearable sensors optimized for step counting may underrepresent activities with minimal foot movement. Older adults’ engagement in low-intensity or incidental activities, such as light housework, may be harder to accurately self-report or detect with wearables, further contributing to the observed discrepancies [64,82,83,84].

These findings suggest that future studies would benefit from integrating self-reports with wearable data to mitigate bias and achieve a more holistic understanding of PA behaviors [85].

Analysis of activity intensity revealed that 51.4% of PA was classified as low-intensity, with moderate-intensity PA contributing 46.2% and vigorous activity accounting for only 2.4%. These findings align with the existing literature indicating that older adults predominantly engage in light activities corresponding to their functional capacities and exertion thresholds [27,86,87]. Martin et al. reported that levels of light activity remained consistent among older women (approximately 30%). In contrast, older men demonstrated more daytime sedentary minutes (~3), fewer daytime light minutes (~4), and more MVPA minutes (~1) than women, with activity declining sharply by early evening [88]. Similarly, Copeland and Esliger found that 90% of activity time in older adults was spent in low-intensity activities, with light activity averaging 13.8–13.9 h per day and MVPA accounting for 68 min, primarily occurring in the morning and as sporadic short bouts. Notably, their findings also highlighted significantly lower PA levels in the evening compared with morning and afternoon [89].

Importantly, PA levels remained stable over the 14-day observation period, with activity peaking at midday and declining in the evening [90,91]. This temporal stability reflects the influence of circadian rhythms and habitual routines, as supported by prior research on biological rhythms and daily activity patterns [92,93,94]. However, the minimal contribution of vigorous activity underscores the need for tailored interventions to incorporate structured, higher-intensity activities into older adults’ daily routines. Strategies to address this gap include the use of behavioral change techniques such as personalized goal setting with gradually increasing intensity and progressive challenges to promote confidence and safety [95,96]. Additionally, integrating high-intensity interval training (HIIT) protocols that are adaptable to individual capabilities may offer a feasible approach to improving cardiovascular and muscular fitness in older adults [97,98]. Furthermore, social engagement can play a pivotal role in encouraging participation in higher-intensity activities [99,100]. Group-based programs and buddy systems have been shown to increase motivation and adherence among older populations [101]. Incorporating gamification elements such as point systems or virtual challenges into technology-based interventions can also provide incentives for sustained engagement in vigorous PA [102,103]. Moreover, the predominance of low-intensity PA highlights a critical gap in engaging older adults in moderate-to-vigorous physical activity (MVPA) [104,105,106]. While light PA is valuable for maintaining mobility and functional independence, this limited participation in MVPA is concerning, given its established benefits for cardiovascular health, cognitive function, and overall well-being [107,108,109]. The observed midday peak in activity represents an optimal window for introducing structured programs to encourage MVPA. Leveraging this natural engagement period could enhance adherence and maximize health outcomes, addressing the disparity between current activity levels and WHO guidelines [62].

Adherence to EMA protocols was relatively high through the follow-up period, with an average completion rate of 67.2% across all time points. This finding highlights the feasibility of EMA as a method for collecting real-time behavioral data in older populations [110]. However, the decline in adherence and motivation during evening assessments suggests potential psychological fatigue or loss of motivation [36]. Adaptive EMA designs, including shorter prompts and engaging elements such as gamification, could mitigate these challenges and improve participant engagement during times of reduced motivation [111].

This study offers several significant strengths and contributions, advancing research on PA behaviors among older adults. Integrating EMA and wearable data provided a multidimensional and ecologically valid perspective, capturing real-time behaviors, psychological states, and contextual factors. This dual approach addresses a critical gap in traditional cross-sectional or self-reported studies by enabling the identification of temporal patterns and within-person variations.

Sustained engagement over the 14-day period, even in real-world, unsupervised settings, demonstrates the practicality of combining wearable technology and EMA for this demographic. This finding highlights the approach’s scalability for use in larger and more diverse populations, particularly in community-based research and interventions [110].

However, an important finding of this study was the absence of difference, both in terms of PA monitoring and EMA responses, when comparing the 14 days of follow-up and shorter random periods of time. Given the fact that we found statistically significant differences in terms of numbers of steps on weekdays and at weekends, we recommend opting for a follow-up duration of one week to be able to capture sufficient data while maintaining high levels of motivation and participation from participants.

Another notable strength of the study is its ecological validity. By capturing behaviors in natural environments, the methodology minimized biases associated with artificial or clinical settings, providing a more accurate representation of real-world activity patterns. Using EMA to assess psychological states in real time complemented wearable data, offering a dynamic understanding of how contextual and temporal factors shaped PA behaviors. The temporal stability observed in PA patterns—aligned with established circadian rhythms and habitual routines—further validates the robustness of the data collection methods. This consistency not only strengthens the reliability of the findings but also highlights strategic opportunities for intervention.

Finally, the study’s focus on older adults—a group often underrepresented in digital health research—stands out as a critical contribution. By tailoring the methodology to this demographic and demonstrating its feasibility, the research lays a foundation for addressing the unique barriers and facilitators of PA in later life [112,113].

Despite its contributions, this study has several limitations that warrant careful consideration and provide valuable direction for future research. First, the sample primarily consisted of physically active and digitally literate older adults, limiting the generalizability of the findings when considering more sedentary or less technologically proficient populations. This selection bias may have resulted in overestimation of adherence rates and engagement with digital tools. To mitigate this issue, future studies should implement stratified sampling strategies to ensure the inclusion of participants with varying levels of PA, digital literacy, and socioeconomic backgrounds [114]. Second, a relatively short 14-day observation period, while sufficient to capture stable PA patterns, does not account for seasonal or lifestyle variations that may influence activity levels over extended periods. To overcome this limitation, future research should extend the monitoring duration to include multiple seasons and key lifestyle events. Additionally, periodic assessments at different intervals (e.g., every six months) could provide a broader understanding of how activity patterns evolve over time and under varying conditions. Therefore, we have planned a six-month follow-up study.

Future research must prioritize methodological adaptations and recruitment strategies that ensure greater inclusivity and external validity [114,115]. Recruitment efforts should target more diverse populations that encompass a range of digital literacy, physical functioning, and baseline activity levels [116]. Additionally, refining EMA protocols is essential for enhancing participant engagement and data fidelity [117,118]. Adaptive EMA designs that account for diurnal variations in motivation and fatigue could include personalized timing of prompts, shorter response formats during evening periods, and optional rest days in extended observation protocols. Gamification elements and interactive feedback mechanisms integrated within EMA platforms may further sustain participant interest and reduce drop-off rates over prolonged studies [119,120,121]. These refinements would improve the feasibility of implementing EMA in broader and less digitally proficient populations.

In the next phase, future research should focus on identifying patterns of PA in relation to environmental and situational factors, utilizing more detailed analyses and advanced methodologies [122,123]. Enhancing data resolution, such as examining shorter time intervals, can provide deeper insights into daily and weekly PA variations. Advanced analytical approaches, including computational and statistical techniques, can be used to identify patterns and classifications within diverse datasets [124,125,126]. Exploring innovative strategies, such as clustering methods or predictive algorithms, may further enhance the ability to detect key trends and inform targeted interventions. Integrating wearable data into JITAIs presents a promising avenue for improving the efficacy of PA promotion strategies. Wearable devices can provide real-time data on activity levels, sleep patterns, and physiological metrics, enabling the development of tailored interventions that respond dynamically to individual needs. For example, JITAIs could deliver personalized reminders, motivational messages, or activity suggestions during periods of inactivity or when participants are most likely to engage in moderate-to-vigorous physical activity [127,128,129]. This adaptive approach could optimize the effectiveness of intervention by aligning strategies with individual behavioral patterns and contextual factors. To support scalability, it is imperative to develop robust automated data-processing pipelines and analytical frameworks. These tools would enable efficient handling of the large volumes of data generated by wearable devices and EMA, facilitating timely and actionable insights [130,131].

## 5. Conclusions

This study reinforces the value of integrating EMA and wearable data to optimize digital phenotyping methodologies for assessing PA patterns among older adults. By merging real-time subjective responses with objective activity measures, this research offers a nuanced understanding of PA behaviors, psychological states, and contextual factors. These findings not only demonstrate the feasibility and acceptability of this approach in an older population but also highlight its potential to identify actionable insights, such as strategic windows for promoting MVPA. This work contributes to advancing digital phenotyping, offering a scalable and adaptable methodology that holds promise for improving PA interventions tailored to the unique needs of older adults.

## Figures and Tables

**Figure 1 sensors-25-00858-f001:**
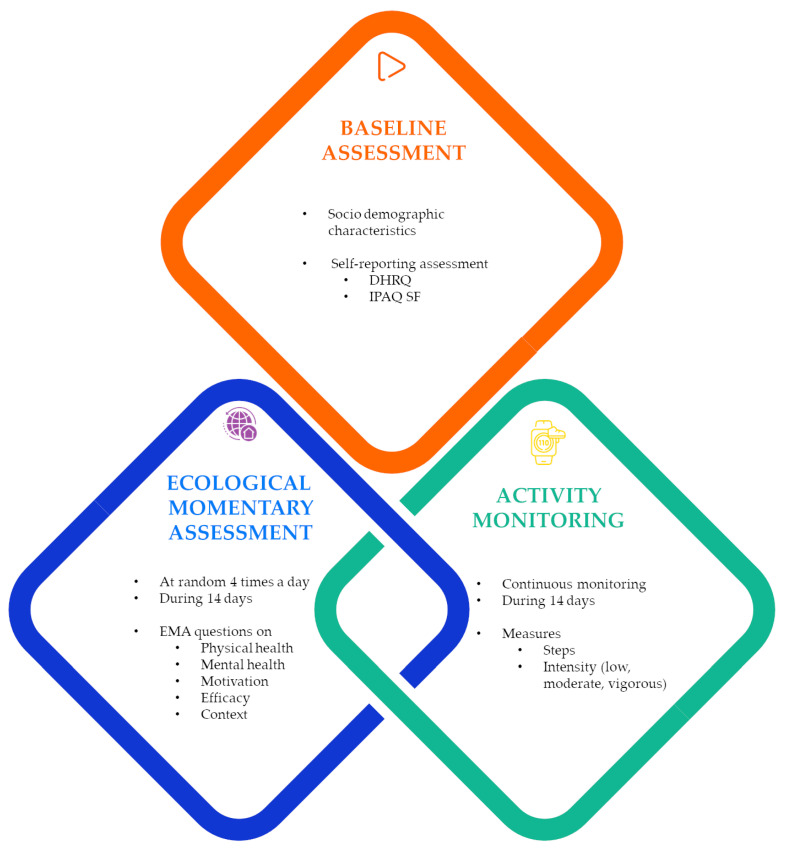
Design of the two-week prospective observational study.

**Figure 2 sensors-25-00858-f002:**
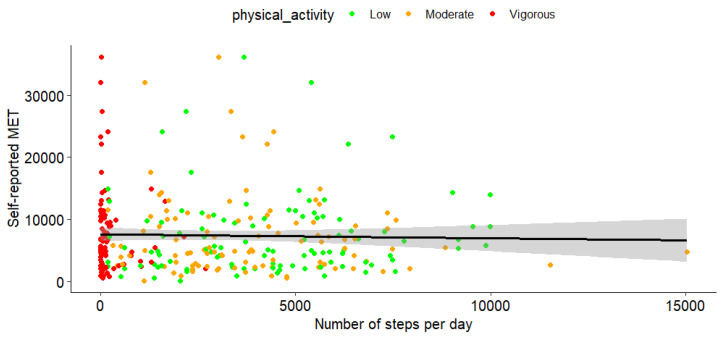
Relationship between self-reported PA level (MET using the IPAQ Short Version) and quantified PA (number of steps per day at various intensity levels). The black line represents the regression trend line, indicating the general relationship between the two variables. The gray shaded area around the black line represents the 95% confidence interval (CI), showing with 95% confidence the range within which the true regression line is likely to lie. The points are color-coded based on PA intensity levels: low (green), moderate (orange), and vigorous (red).

**Figure 3 sensors-25-00858-f003:**
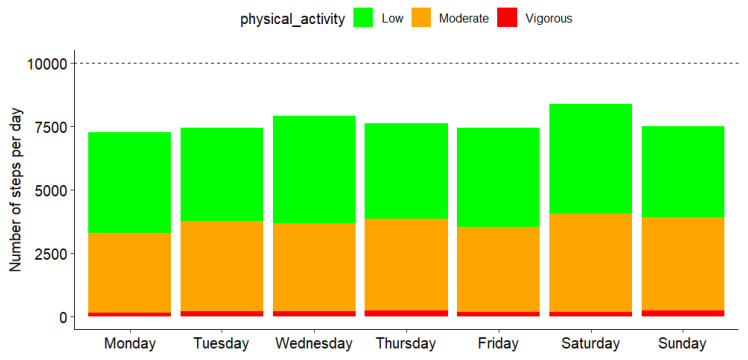
Aggregated average number of steps according to time of day. The dashed horizontal black line indicates WHO’s recommendation.

**Figure 4 sensors-25-00858-f004:**
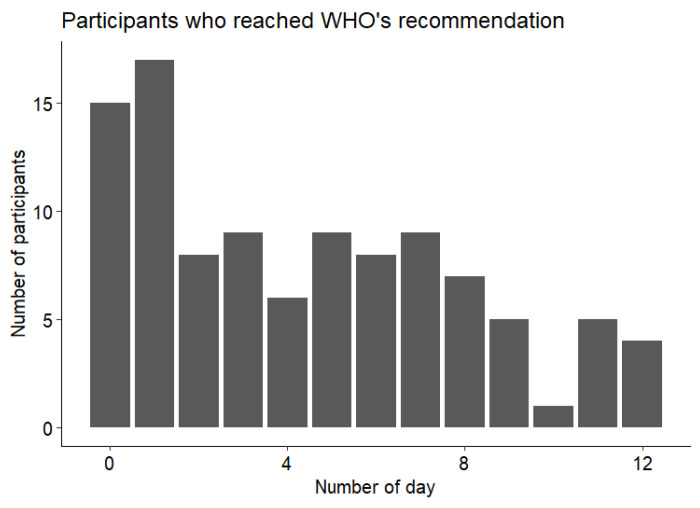
Distribution of the participants according to the number of days where the WHO’s recommendations of 10,000 steps were reached.

**Figure 5 sensors-25-00858-f005:**
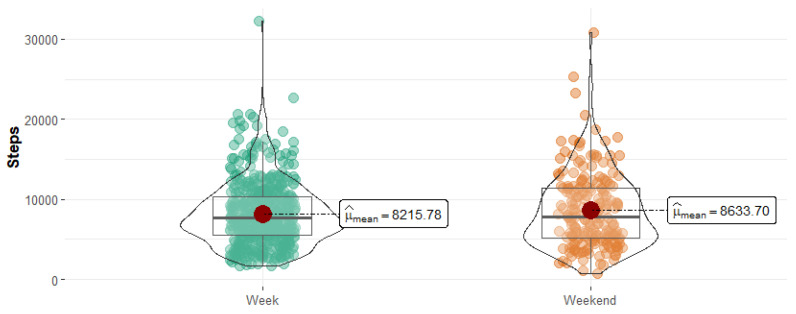
Variations in physical activity patterns: a comparison of weekday and weekend step counts.

**Figure 6 sensors-25-00858-f006:**
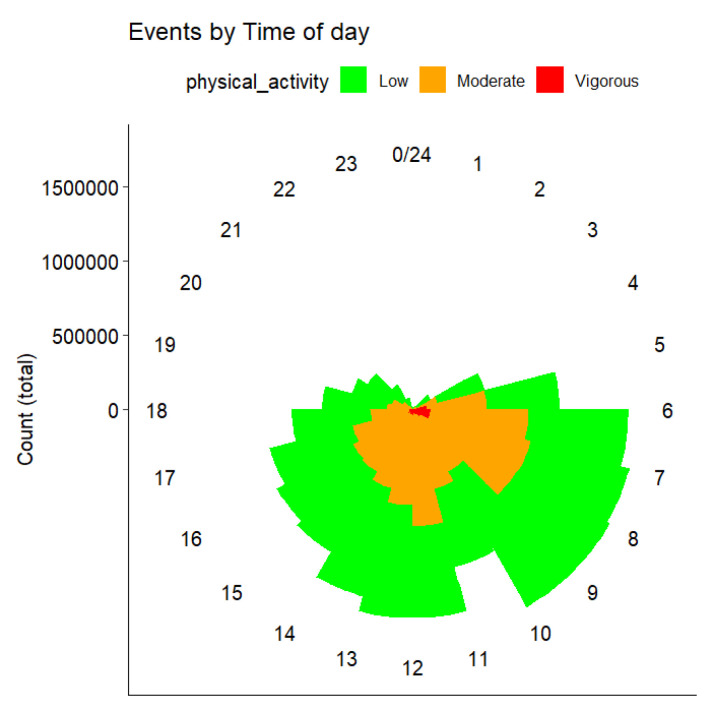
Aggregated physical activity events by time of day and intensity level.

**Figure 7 sensors-25-00858-f007:**
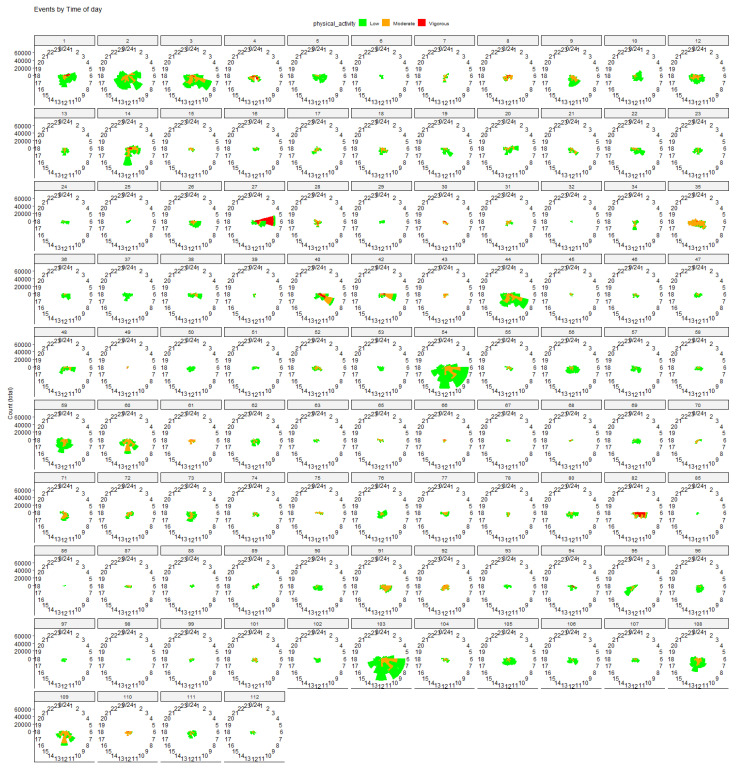
Daily PA patterns across participants (n = 108), categorized by intensity level and time of day.

**Figure 8 sensors-25-00858-f008:**
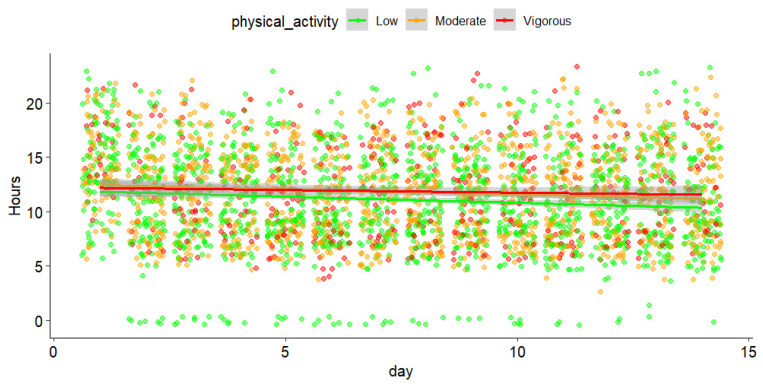
Temporal patterns of PA intensity levels over a 14-day period. The scatter points represent individual data points for PA intensity, color-coded by level: green for low, orange for moderate, and red for vigorous activity. The green, orange, and red lines represent the average trends of low, moderate, and vigorous activity levels, respectively, across the hours of the day and the 14-day observation period. The gray zones surrounding these lines indicate the 95% confidence intervals (CI), showing the variability in activity intensity trends across participants.

**Figure 9 sensors-25-00858-f009:**
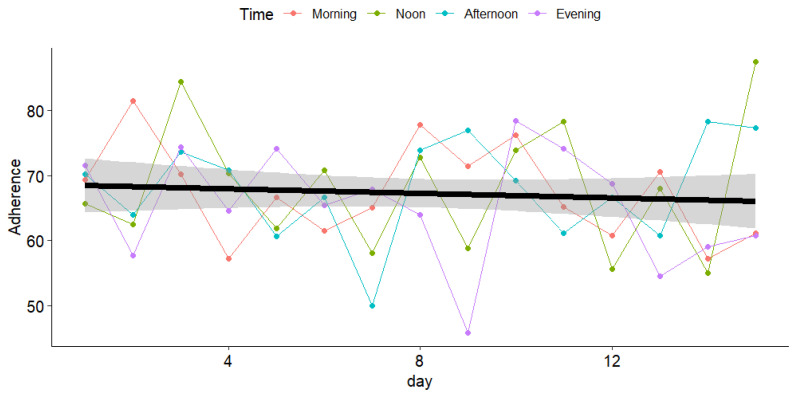
Adherence to EMA over time, stratified by time of day. The colored lines (pink for morning, green for noon, blue for afternoon, and purple for evening) represent adherence trends at different times of the day throughout the 14-day observation period. The black line represents the overall adherence trend across all time periods, while the gray shaded area indicates the 95% confidence interval (CI) for this overall trend.

**Figure 10 sensors-25-00858-f010:**
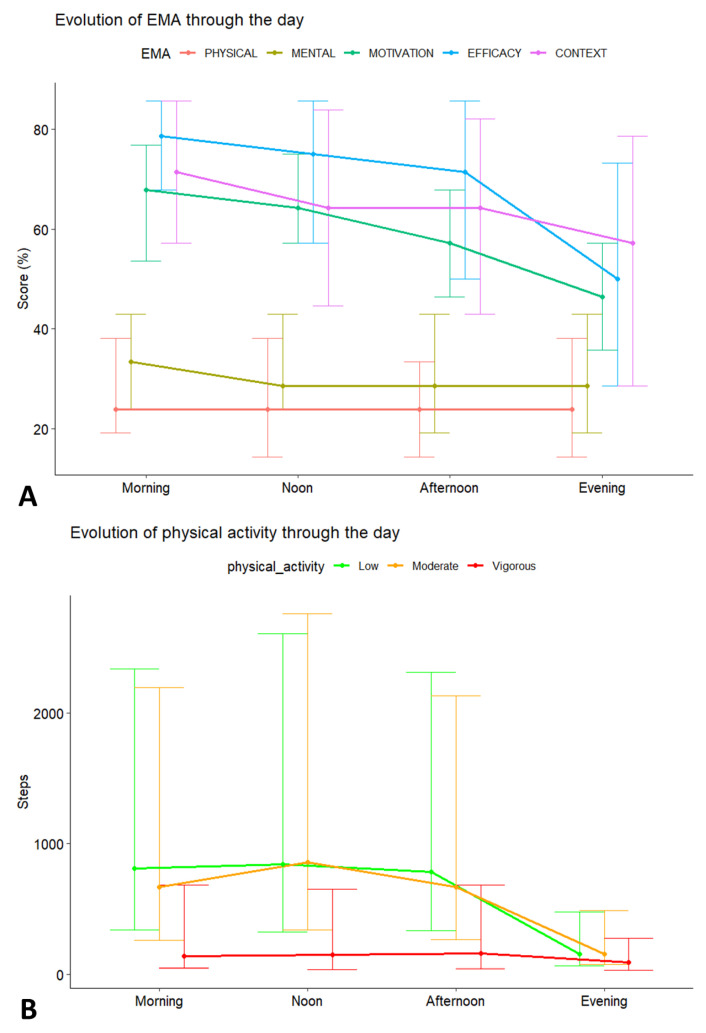
Evolution of EMA (**A**) and PA (**B**) throughout the different time periods.

**Figure 11 sensors-25-00858-f011:**
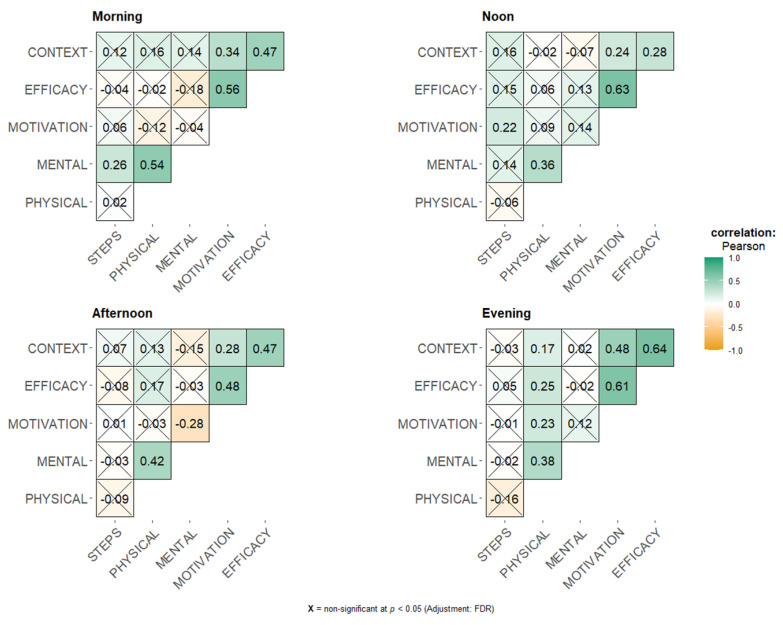
Temporal dynamics of psychological and physical correlations across time stamps.

**Figure 12 sensors-25-00858-f012:**
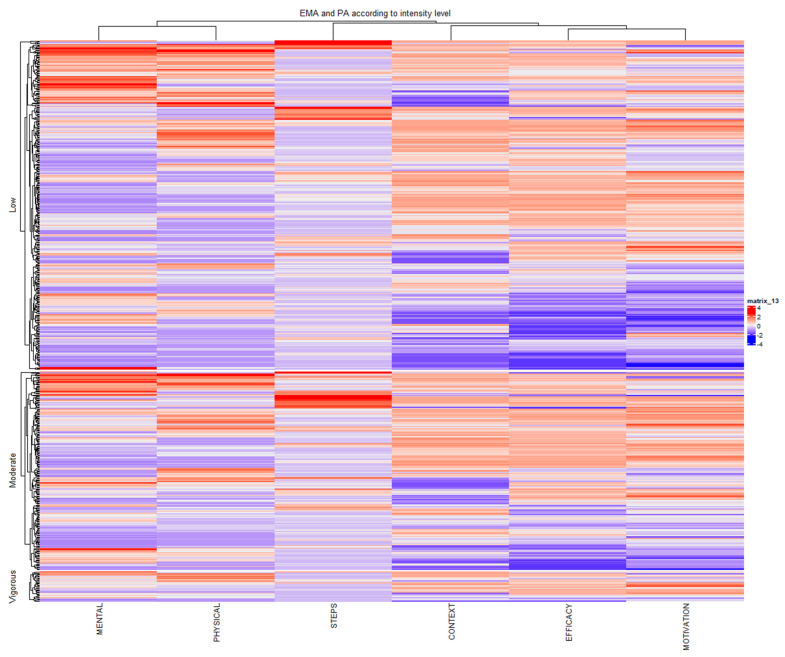
Heatmap analysis of EMA categories and PA across activity levels. The heatmap values range from −4 to +4, representing normalized z-scores of the variables analyzed. Positive values (shades of red) indicate above-average levels relative to the dataset, while negative values (shades of blue) indicate below-average levels. The intensity of the color corresponds to the magnitude of the z-score, with deeper shades reflecting greater deviations from the mean.

**Figure 13 sensors-25-00858-f013:**
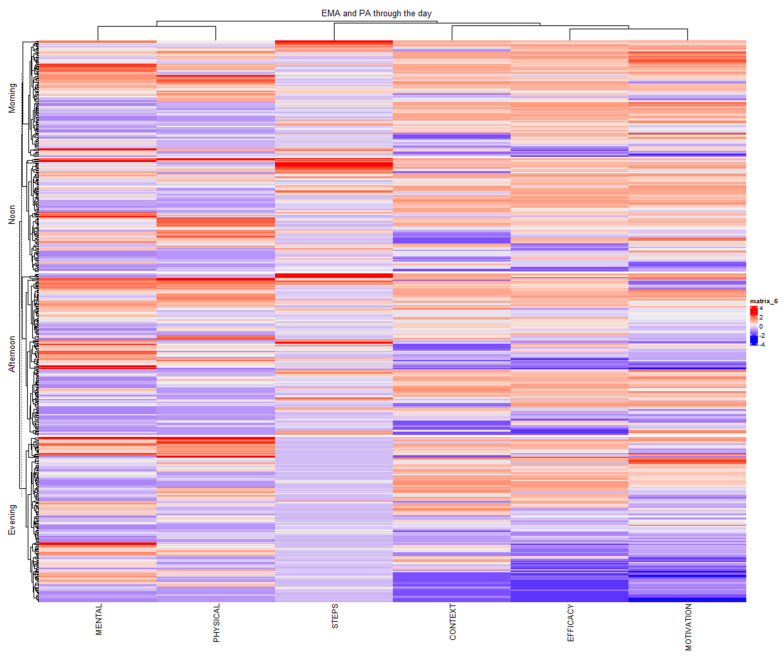
Heatmap analysis of EMA categories and PA across the day. The heatmap values range from −4 to +4, representing normalized z-scores of the variables analyzed. Positive values (shades of red) indicate above-average levels relative to the dataset, while negative values (shades of blue) indicate below-average levels. The intensity of the color corresponds to the magnitude of the z-score, with deeper shades reflecting greater deviations from the mean.

**Table 1 sensors-25-00858-t001:** Sociodemographic characteristics of the participants.

Variable	Value (n = 108)
Demographics	
Age (years)	69 [65–87]
Sex (female)	60 (55.6)
BMI (kg/m^2^)	26.3 (19–42)
Marital status	
Single	8 (7.4)
Living together	9 (8.4)
Married	78 (72.2)
Divorced	8 (7.4)
Widow	5 (4.6)
Educational level	
Primary school	3 (2.8)
Secondary education	28 (25.9)
High school	52 (48.2)
University	23 (21.3)
PhD	2 (1.8)
Living situation	
Living with partner	85 (78.7)
Living alone	20 (18.6)
Living with children	1 (0.9)
Other	2 (1.8)
Fall incidence (yes)	18 (16.7)
Participant-reported outcomes	
IPAQ-SF (total METmin/week)	5154 [99–64,848]
Low	1 (0.7)
Moderate	30 (28)
High	77 (71.3)
Digital literacy, total score (75)	60 ± 8
Usage (20)	16 [10–20]
Skills (25)	22 [12–25]
Literacy (15)	12 [7–15]
Health literacy (15)	10 [3–15]
Learnability (25)	19 [5–25]

Data are expressed as mean ± SD, median [min–max], or n (percentages). Abbreviation: IPAQ-SF: International Physical Activity Questionnaire—short form.

**Table 2 sensors-25-00858-t002:** Effect of follow-up duration on PA and EMA categories across various time periods.

Category	Outcome	Time	Complete (14 Days)	3 Days	7 Days	10 Days	12 Days	*p*-Value
PA	Low	Morning	810 (1187)	799 (1140)	818 (1195)	818 (1210)	817 (1214)	0.58
		Noon	844 (1440)	905 (1377)	874 (1380)	843 (1436)	830 (1465)	0.86
		Afternoon	784 (1191)	824 (1305)	820 (1205)	791 (1212)	792 (1216)	0.84
		Evening	153 (258)	189 (264)	162 (242)	153 (246)	150 (253)	0.70
	Moderate	Morning	668 (1266)	695 (1157)	624 (1172)	679 (1276)	684 (1287)	0.55
		Noon	860 (1152)	889 (1517)	829 (1578)	880 (1624)	905 (1628)	0.76
		Afternoon	667 (1200)	631 (1264)	648 (1260)	678 (1274)	672 (1208)	0.48
		Evening	154 (258)	142 (312)	152 (268)	156 (266)	155 (269)	0.66
	Vigorous	Morning	139 (495)	75 (437)	104 (349)	126 (410)	139 (436)	0.22
		Noon	139 (495)	322 (534)	149 (478)	146 (465)	155 (459)	0.45
		Afternoon	158 (495)	260 (395)	260 (610)	218 (496)	200 (491)	0.67
		Evening	90 (152)	120 (157)	116 (161)	78 (179)	80 (57)	0.35
EMA	Physical	Morning	23.8 (23.8)	23.8 (23.8)	23.8 (19.0)	23.8 (19.0)	23.8 (19.0)	0.96
		Noon	23.8 (19.0)	28.6 (21.4)	23.8 (17.9)	23.8 (19.0)	23.8 (19.0)	0.87
		Afternoon	23.8 (23.8)	19.0 (17.9)	23.8 (19.0)	23.8 (19.0)	23.8 (19.0)	0.65
		Evening	23.8 (19.0)	23.8 (26.2)	23.8 (23.8)	23.8 (23.8)	23.8 (23.8)	0.84
	Mental	Morning	33.3 (19.0)	28.6 (19.0)	28.6 (16.7)	28.6 (19.0)	28.6 (14.3)	0.88
		Noon	28.6 (19.0)	28.6 (16.7)	28.6 (19.0)	28.6 (23.8)	28.6 (14.3)	0.99
		Afternoon	28.6 (23.8)	31.0 (25.0)	28.6 (22.6)	28.6 (23.8)	28.6 (23.8)	0.87
		Evening	28.6 (23.8)	28.6 (21.4)	28.6 (23.8)	28.6 (23.8)	28.6 (23.8)	0.94
	Motivation	Morning	67.9 (23.2)	67.9 (21.4)	67.9 (23.2)	67.9 (21.4)	67.9 (23.2)	0.92
		Noon	64.3 (17.9)	60.7 (8.9-	64.3 (14.3)	64.3 (10.7)	64.3 (14.3)	0.86
		Afternoon	57.1 (21.4)	55.4 (16.1)	55.4 (14.3)	57.1 (21.4)	55.4 (21.4)	0.89
		Evening	46.4 (21.4)	39.3 (16.1)	42.9 (21.4)	42.9 (21.4)	42.9 (21.4)	0.84
	Efficacy	Morning	78.6 (17.9)	78.6 (14.3)	85.7 (17.9)	78.6 (21.4)	78.6 (21.4)	0.84
		Noon	75.0 (28.6)	64.3 (39.3)	75.0 (28.6)	71.4 (28.6)	71.4 (26.8)	0.86
		Afternoon	71.4 (35.7)	60.7 (35.7)	57.1 (35.7)	71.4 (35.7)	71.4 (35.7)	0.82
		Evening	50 (44.6)	42.9 (67.9)	50 (50)	50 (42.9)	50 (42.9)	0.67
	Context	Morning	71.4 (28.6)	64.3 (21.4)	71.4 (21.4)	71.4 (28.6)	71.4 (25.0)	0.89
		Noon	64.3 (39.3)	64.3 (25.0)	67.9 (41.1)	64.3 (35.7)	64.3 (28.6)	0.78
		Afternoon	64.3 (39.3)	67.9 (32.1)	57.1 (35.7)	64.3 (35.7)	64.3 (35.7)	0.64
		Evening	57.1 (50)	57.1 (53.6)	57.1 (57.1)	57.1 (50)	57.1 (50)	0.87

**Table 3 sensors-25-00858-t003:** Summary of Findings Addressing Research Goals in Digital Phenotyping for Physical Activity Monitoring.

Research Goal	Findings
1. Assess the interplay between PA patterns and psychosocial determinants across different times of the day and intensities.	Temporal patterns: Most PA occurred in the mornings (11:00–12:00 was the most active hour). Low-intensity activity was the most common (51.4%), followed by moderate (46.2%), and vigorous (2.4%). PA declined significantly in the evenings (β = −2.54, SE = 0.27, *p* < 0.001). Daily variability: PA was higher on weekdays compared with weekends (*p* < 0.001), but no differences were observed across individual weekdays (*p* = 0.47). Correlations: PA showed weak but significant correlations with motivation (R = 0.16, *p* = 0.001), context (R = 0.11, *p* = 0.007), efficacy (R = 0.10, *p* = 0.006), and mental health (R = 0.09, *p* = 0.031). No correlation was observed with physical health (R = −0.05, *p* = 0.31). Intensity-specific correlations: At low PA intensity, motivation (R = 0.20, *p* = 0.003), efficacy (R = 0.14, *p* = 0.013), and context (R = 0.13, *p* = 0.019) were positively correlated with PA. Cluster analysis: Self-efficacy and motivation were consistently clustered, suggesting a strong relationship, while context was inversely related to these factors in the moderate and low PA clusters. Vigorous PA showed localized clustering with physical health and motivation.
2. Evaluate the feasibility of these methodologies, including participant adherence within a representative sample of older adults.	Adherence: Overall adherence to EMA was 67.2% (SD = 8.4), with similar rates across all time periods: morning (67.4%), noon (68.2%), afternoon (68.0%), and evening (65.4%). No significant differences were found across time periods (*p* = 0.82). Consistency of Engagement: Temporal and intraday variability of PA and EMA data indicated that participants remained engaged throughout the study period, with no major drop-offs. Activity levels: In total, 71.3% of participants reported high PA levels (median total MET-min/week: 5154, IQR: 7331), 28% moderate, and 0.7% low, demonstrating a representative and physically active sample.
3. Explore optimal durations for data collection to ensure participant engagement while capturing meaningful behavioral trends.	PA stability across durations: PA levels showed no significant differences across follow-up durations (1, 3, 7, 10, 12, and 14 days) for low (*p* = 0.58), moderate (*p* = 0.55), or vigorous intensities (*p* = 0.22). EMA stability across durations: EMA responses were consistent across durations for all categories: physical health (*p* = 0.96), mental health (*p* = 0.88), motivation (*p* = 0.92), efficacy (*p* = 0.84), and context (*p* = 0.89). Optimal duration: Shorter durations, such as 7 days, provided insights similar to those gained from the full 14-day period, making this a feasible alternative while reducing participant burden. Consistency in Temporal Patterns: The most active hour (11:00–12:00) and diurnal trends in PA (higher in the morning, declining in the evening) were consistent across all durations. EMA responses also demonstrated similar temporal trends, with efficacy and motivation highest in the morning and declining throughout the day.

## Data Availability

The data presented in this study are available on request from the corresponding author due to privacy, legal or ethical reasons.

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
