# Peer review of "From Steps to Context: Optimizing Digital Phenotyping for Physical Activity Monitoring in Older Adults by Integrating Wearable Data and Ecological Momentary Assessment"

_sensors, 2025, doi:10.3390/s25030858_

Round 1

Reviewer 1 Report

Comments and Suggestions for Authors

In this work, the authors proposed optimizing digital phenotyping strategies for assessing physical activity (PA) patterns in older adults by integrating ecological momentary assessment (EMA) and continuous wearable sensor data. The findings indicated that a combined approach of wearable sensors and EMA is feasible for evaluating PA, with notable associations between low-intensity PA and psychological factors like motivation and self-efficacy. However, discrepancies were noted between self-reported and sensor-measured PA, suggesting the complementary value of both data types. This work is of interest, however, there are some concerns of this reviewer to be addressed. Please find below my comments:

-            The study found no correlation between self-reported MET values and wearable step data (R = -0.026, p = 0.65). Could you provide more detail on the potential reasons behind this lack of correlation, especially in light of the methodological differences between subjective and objective measures?

-            The temporal patterns of PA demonstrated that low and moderate intensity were more prevalent, while vigorous activity was limited. Could you discuss any specific strategies you employed to encourage higher intensity activity, and how these strategies could be tailored to older adults in future interventions?

-            The study utilized the Garmin Vivo 5 device for continuous monitoring of PA. Can you elaborate on the accuracy of this wearable device in measuring activity levels in older adults, and whether you considered using other devices with more advanced capabilities in terms of physiological metrics like heart rate variability or oxygen saturation?

-            EMA adherence was noted to decline slightly in the evening. Did you explore any specific factors that could explain this drop in response rates, such as fatigue or social engagement, and how might you optimize EMA designs to improve engagement during these periods?

-            In terms of activity intensity distribution, your study showed that only 2.4% of steps were classified as vigorous. What specific interventions do you recommend to increase participation in vigorous PA, considering that older adults might face physical or psychological barriers to higher intensity exercises?

-            Your analysis used time-series methods to explore temporal PA patterns. How do you plan to improve the granularity of this temporal data to capture intra-day variations and potential changes in PA that could occur under varying environmental or situational contexts?

Comments on the Quality of English Language

Acceptable

Author Response

Dear reviewer

Thank you for your feedback and suggestions on our manuscript. We have carefully addressed each of your comments, and you will find our detailed responses in the attached document. The revised manuscript with track changes has also been attached for your review.

We greatly appreciate the time and effort you have taken to provide constructive feedback.

Best regards

Kim

Reviewer 2 Report

Comments and Suggestions for Authors

From Steps to Context: Optimizing Digital Phenotyping for Physical Activity Monitoring in Older Adults by Integrating Wearable Data and Ecological Momentary Assessment Authors

This study aimed to optimize digital phenotyping strategies for assessing patterns in older adults by integrating ecological momentary assessment (EMA) and continuous wearable sensor data collection.

However, I have some major concerns as follows.

1.     The section literature is missing. Some literature is descripted in section introduction. However, the discussion of literature is few and can’t address the contribution of literature gap.

2.     The secttion of method, especially study setting and design, is too rough to justify the design and procedure of this study.

      3.     It’s notable that the limitations of sample selection and short 14-day observation period weaken this paper’s contributions and validation.

Author Response

(The authors gave the same response as above.)

Reviewer 3 Report

Comments and Suggestions for Authors

Main Issues:

1. In-depth analysis: The paper conducts statistical analyses from multiple dimensions of the collected data, but the purpose of these analyses is not well articulated. What problem is this study attempting to solve? The findings are fragmented and fail to converge into a solid, cohesive result.

2. Sample Size and Duration: Why were 108 participants chosen, and why was a 2-week trial period used? Are there supporting references to justify these choices? Is this sample size and duration sufficient to conduct a robust statistical study? The authors need to provide clear rationales.

3. Integration of Methods: Although the authors aimed to integrate multiple evaluation methods, particularly EMA responses and PA monitoring, the key findings predominantly rely on PA monitoring. For instance, Figures 3-8 primarily present results based on PA monitoring. Figures 2, 9-13 involve comparisons or correlations between PA and EMA data. However, no integration or optimization results are demonstrated, which undermines the study’s main objective.

Detailed Issues:

1. Figure 2: The number of steps per day measures steps, while metabolic equivalents (MET) measure walking cadence/intensity level. Why are these two metrics compared when presenting relationship between self-reported results and sensor measurements? If there is a conversion between walking cadence and steps, consider using a single metric for consistency.

2. Line 16: Does the set of four randomized prompts remain the same for all participants? If they differ, how might this variability impact the participants’ PA motivation?

3. Lines 194–195: The sentence, “To latter ease …,” is unclear and needs to be rewritten for better comprehension.

4. Abbreviations: Provide the full term when an abbreviation is used for the first time.

5. Line 235: Clarify the term “a random sliding window.” How is this window generated, and what criteria are used for its randomness?

6. Figures 2, 8, and 9: The meaning of the plots needs to be explained in detail. What do the black line and the surrounding gray area represent?

7. Figures 12 and 13: Explain the significance of the heatmap values ranging from -4 to 4. Additionally, the y-axis labels are not sequentially ordered and need correction.

Author Response

Dear reviewer

Thank you for your feedback and suggestions on our manuscript. We have carefully addressed all your comments, and you will find our detailed responses in the attached document. Additionally, the revised manuscript with track changes has also been uploaded on the plafform.

We greatly appreciate the time and effort you have taken to provide constructive feedback.

Best regards

Kim

Round 2

Reviewer 1 Report

Comments and Suggestions for Authors

Thank you for addressing the comments of this reviewer. I have no more comments.

Author Response

Dear reviewer

Thank you for your feedback and for taking the time to review our manuscript. We greatly your thorough assessment and are pleased that the revisions meet your expectations.

Best regards,

Kim

Reviewer 2 Report

Comments and Suggestions for Authors

Review Report Form

[Sensors] Manuscript ID: sensors-3398258 (2nd)

Title

From Steps to Context: Optimizing Digital Phenotyping for Physical Activity Monitoring in Older Adults by Integrating Wearable Data and Ecological Momentary AssessmentAuthors

In this revised version, the authors have revised and enhanced relevant references in the introduction and clarified the research questions by point to point to contextualize the study's objectives and contributions better. The Methods section was improved to provide greater clarity and explanation of the figires. The revised results and conclusion are more clear and fair.

Author Response

Dear reviewer

Thank you for your positive feedback on our revisions. We are pleased that the revisions have met your expectations.

We greatly appreciate your time and effort in reviewing our manuscript.

Best regards

Kim

Reviewer 3 Report

Comments and Suggestions for Authors

There is a typo in Line 506, my -> may

The revised manuscript demonstrates substantial improvements in both the design of the experimental methods and the clarity and depth of the results' description. The authors have effectively addressed the previous concerns, and the revisions have strengthened the overall quality of the study. The identification of certain patterns in this study provides some insights into the topic. However, conducting a more in-depth analysis of these patterns would provide additional context and elevate the paper's overall significance and impact. Additionally, comparing these findings with existing literature could help position the study within the broader research landscape, highlighting its unique contributions.

Author Response

Dear reviewer

Thank you for your valuable feedback and your positive assessment of our revisions.

Comment 1: There is a typo in Line 506, my -> may

Reply 1: Thank you for pointing out the typo. We have corrected "my" to "may" (Line 512)

Comment 2: The revised manuscript demonstrates substantial improvements in both the design of the experimental methods and the clarity and depth of the results' description. The authors have effectively addressed the previous concerns, and the revisions have strengthened the overall quality of the study. The identification of certain patterns in this study provides some insights into the topic. However, conducting a more in-depth analysis of these patterns would provide additional context and elevate the paper's overall significance and impact. Additionally, comparing these findings with existing literature could help position the study within the broader research landscape, highlighting its unique contributions.

Reply 2:  We appreciate your suggestion to conduct a more in-depth analysis of the identified patterns and to compare our findings with existing literature. We acknowledge that such an analysis has the potential to further enhance the impact of our study. However, at this moment, it is not entirely clear which specific additional analyses you are referring to. Given the limited timeframe available (one day for revisions), it is unfortunately not feasible for us to conduct a comprehensive extension of the analysis. That said, if you have concrete suggestions for targeted follow-up analyses, we would be happy to consider them for future phases of this study. To address your valuable input within the given constraints, we have expanded the discussion section to provide additional context for our findings. We enriched it with relevant literature to better support our conclusions on the PA patterns.

Referral in manuscript: Discussion | p23 | Line 534 – 542 and Line 546

Kind regards,

Kim Daniels
